# Community Perception and Attitude towards COVID-19 Vaccination for Children in Saudi Arabia

**DOI:** 10.3390/vaccines11020250

**Published:** 2023-01-22

**Authors:** Waddah M. Alalmaei Asiri, Ayed A. Shati, Syed E. Mahmood, Saleh M. Al-Qahtani, Youssef A. Alqahtani, Raghad M. Alhussain, Noura A. Alshehri

**Affiliations:** 1Department of Medicine, College of Medicine, King Khalid University, Abha 62529, Saudi Arabia; 2Department of Child Health, College of Medicine, King Khalid University, Abha 62529, Saudi Arabia; 3Family and Community Medicine Department, College of Medicine, King Khalid University, Abha 62529, Saudi Arabia; 4College of Medicine, King Khalid University, Abha 62529, Saudi Arabia

**Keywords:** community, perception, attitude, COVID-19, vaccination, children, Saudi Arabia

## Abstract

Introduction: Vaccines are an important part of the COVID-19 pandemic response plan. This study was undertaken to find out the percentage of supporters in the Saudi population for COVID-19 vaccination among children, and to assess the study population’s perceptions towards COVID-19 vaccination among children. Material and Methods: This nationwide study adopted a cross-sectional survey of adult participants, conducted by trained medical students. The anonymous questionnaire was published on social media tools. Statistical analysis was conducted using two-tailed tests. Results: Six hundred and twenty (620) participants were recruited for this study. Nearly 17.0% of participants had a chronic health problem/comorbidity. About 28.7% of the study participants reported having COVID-19 infection. The COVID-19 vaccine was received among 94.7% of the study respondents. The majority of the vaccination supporters (89.0%) wanted to get the third dose. There was a statistically significant association between the participant’s attitudes towards getting vaccinated themselves, and their attitude towards children’s vaccination. Vaccine newness, as a reason for hesitating to get vaccinated, was most reported among non-vaccine supporters. False religious beliefs were found to significantly influence the opposing attitude towards children’s vaccination. Conclusion: Health professionals and policy makers should implement and support strategies to ensure children are vaccinated against COVID-19. They also need to educate parents and families regarding the importance of vaccination against COVID-19.

## 1. Introduction

In Saudi Arabia, as of 28 October 2022, there have been 821,246 confirmed cases of COVID-19 with 9400 deaths, reported to the WHO. A total of 67,839,503 vaccine doses have been administered in the entire Kingdom of Saudi Arabia [1]. Vaccines are an important part of the COVID-19 pandemic response plan. They represent one of the most successful and cost-efficient public health interventions developed, saving millions of lives every year [2,3,4]. Following the decoding of the SARS-CoV-2 genome sequence in early 2020 [5] and the World Health Organization’s (WHO) declaration of the pandemic in March 2020 [6], scientists and pharmaceutical firms were racing against the clock to create effective vaccines [7]. Around the world, there are now 137 COVID-19 vaccine candidates undergoing clinical trials, and 194 candidates in pre-clinical development [8]. AstraZeneca Oxford, Moderna, Pfizer, and Johnson & Johnson have been approved for use in Saudi Arabia [9]. In Saudi Arabia, Pfizer–BioNTech and Moderna vaccines are currently available and funded for children > 4 years old [10]. According to a 2019 report from the WHO, one of the 10 threats to global health is vaccine hesitancy, which is defined as the reluctance or refusal to vaccinate despite the availability of vaccines [11,12]. There was a positive attitude towards vaccination among the majority of the participants from the Saudi Arabian population, who were willing to contemplate taking the vaccines and have them administered to their children, as recommended by healthcare centers [13]. However, there is a reluctance noticed towards children having the COVID-19 vaccination, worldwide. Some studies in the Kingdom of Saudi Arabia have concluded that vaccine hesitancy towards COVID-19 among parents is a concern, and is likely to influence the COVID-19 vaccination status of their children [14]. Thus, adult vaccine hesitancy impacts childhood uptake. However, further studies are required to investigate the acceptance and hesitancy levels in the Saudi population. In adult SARS-CoV-2 infections, patients with pre-existing underlying comorbidities, such as chronic obstructive pulmonary disease, cardiovascular disease, diabetes, and obesity, are more likely to have severe disease compared to healthy adults [15]. An inconsistency is seen in current findings on the association with comorbidities and pediatric COVID-19 severity [16]. Patients with comorbidities may have conflicting COVID-19 vaccine attitudes. Few studies have specifically explored the issues of COVID-19 vaccine hesitancy among patients with severe comorbid conditions, or strategies to increase acceptance in high-risk populations. [17]. Therefore, this study was undertaken to find out the percentage of supporters and opposers in the Saudi population for COVID-19 vaccination among children to assess the study population’s perceptions towards COVID-19 vaccination among children, and find out the relationship between public opinion towards children’s vaccination and sociodemographic variables.

## 2. Material and Methods

### 2.1. Study Design & Duration

This study was approved by the Research Ethical Committee of the College of Medicine, King Khalid University (REC#2021). This study adopts a cross-sectional survey design of participants nationwide, conducted by trained medical students from August 2021 to October 2021. We invited a random sample from the original list of every mobile phone user to participate. All adult citizens who had mobile phones were included. The research team made the random allocation using an Excel program that generated the required mobile phone lists to eliminate any selection bias and forwarded them to the data collectors. Medical students filled out the answers of the callers in online questionnaires. Two of the authors were responsible for tracking the data collectors and the participants’ responses to assure full compliance with the study methodology.

### 2.2. Population, Inclusion and Exclusion Criteria and Sample Size

The study included adults residing in different geographical locations of Saudi Arabia who consented to participate in the study and were enrolled in the study’s final analysis. The participants were identified randomly from the list of mobile phone users. Persons who declined to consent were aged below 18 years, or spoke a language other than Arabic, and were excluded from the study.

According to Epi info Version [18], a sample size of 384 participants was estimated with a 5% margin of error and a 95% confidence interval. However, the statistical analysis and results were conducted on a sample of 620. A total of 670 people were invited. The response rate was 92.6%.

### 2.3. Data Collection Tool

A cross-sectional web-based study design was employed to gather data about willingness, hesitancy factors, and attitudes towards COVID-19 vaccination in Saudi Arabia. The anonymous questionnaire was published on social media tools. Respondents were encouraged to participate in this study by clarifying the extent of confidentiality of participation, and the importance of this research to society’s health. Data were collected using an adapted and modified questionnaire based on literature and expert opinion. The questionnaire was distributed online to the participants, and comprised closed ended questions made to meet the study objectives. The online questionnaire was prepared in both English and Arabic languages using Google forms, and distributed among participants through social media and e-mail for convenience of data collection, as face-to-face interviews had to be avoided following the social distancing norms set by the government. The questionnaire was translated from English to Arabic (local language) by a bilingual person to enable an easy understanding of the questions, and to avoid any questionnaire bias. Before administration of the final version of the questionnaire, a pretest was performed among 30 random parents in the region to ensure the reliability and applicability of the questionnaire. The results of the piloted study were not included in the final analysis. The face and content validity of the questionnaire was assessed by specialists in the fields of research. Content validity was evaluated by the content validity index, which was 0.82. The reliability of the questionnaire was assessed by using test-retest reliability method. The questionnaire was modified according to the participant’s suggestions, and comments to make it more comprehensive and understandable. The data collection sheet consisted of three parts. The first part included demographic and medical information. The second set of questions focused on participants’ acceptance of the vaccine. Finally, a third set attempted to gauge hesitancy towards the COVID-19 vaccine. The data were collected by research assistants who had a workshop on the datasheet of this study. The research project was explained to the participants during the phone call, and signed consent was sent by email or other means.

The questionnaire of this study includes the following parts:

Part 1: Demographics variables: Age, Gender, Geographic Location, Education, Occupation, Monthly Household Income, Marital Status, and Living Status

Part 2: Co-morbidities, COVID-19, and vaccine history: This part includes five items, assessed by (Yes/No) response. Comorbidity: the presence was mainly dependent on participants’ self-report, including hypertension, heart diseases, diabetes, chronic obstructive pulmonary disease (COPD), asthma, chronic liver disease, chronic kidney disease, malignant tumor, human immunodeficiency virus infection, hematologic disease, and other comorbidities that may have an influence on the illness, including the use of immunosuppressants, tuberculosis, hyperthyroidism, hypothyroidism, and cerebrovascular diseases.

Part 3: Attitudes, beliefs, and emotions towards the COVID-19 vaccine: This scale includes six items. Each item was scored on a 5-point Likert scale from strongly disagree (1) to strongly agree (5); a high score indicates accepting attitudes towards the COVID-19 vaccine.

Five-level Likert scale measurements

**Strongly Disagree****Disagree****Neutral****Agree****Strongly Agree**Item12345

Part 4: Three questions about respondents’ attitude if vaccination became mandatory by the authorities and their attitude towards the third dose of vaccine (Foster), each one assessed by (Yes/No) response.

Vaccination supporters are those who answered yes for children’s vaccination, and the opposers are those who answered no for children’s vaccination.

### 2.4. Data Analysis

The collected data were coded and entered into an excel software (Microsoft office Excel 2010) database. Data were analyzed using Statistical Package for Social Sciences, version 25.0 (SPSS, Inc., Chicago, IL, USA). All statistical analysis was conducted using two-tailed tests. *p* value less than 0.05 was statistically significant. Descriptive analysis was used, and associations between vaccine uptake and demographic variables were studied. Descriptive analysis based on the frequency and percent distribution was conducted for all variables, including participant’s socio-demographic data, medical history, COVID-19 infection, and COVID-19 vaccine intake with received doses. In addition, participants’ reasons for having the vaccine, reasons for hesitancy among those who were not vaccinated, and their attitudes and perceptions regarding COVID-19 vaccine intake were shown in frequency tables. Cross tabulation was used to assess the distribution of study population vaccination and their related bio-demographic data. Relations were tested using the Pearson chi-square test and exact probability test for small frequency distributions. Univariate analysis was used to find the association of opponents of vaccination with demographic characteristics applying a chi-square test at the 5.00% level of significance. Multivariate analysis (logistic regression model) was further used. Regression analysis was conducted using ‘I am an opponent of vaccination in general’ as the dependent variable, and the select socio-demographic variables as independent variables. The variable ‘I am an opponent of vaccination in general’ used was a dichotomous variable.

## 3. Results

### Participants Characteristics

Six hundred and twenty (620) participants were recruited for this study. Socio-demographic characteristics were collected, as presented in Table 1:

Out of the study population, 355 (57.2%) were males, while 265 (42.8%) were females. All age groups had a good representation, with 34.0% in the age group of 18–25 years, 30.0% in the age group of 26–35 years, 10.8% in the age group of 36–40 years, 10.0% in the age group of 41–45 years, and 7.8% aged more than 50 years, while 7.4% of participants were aged 46–50 years. A higher percentage of participants (53.1%) were married, 40.6% were single, 5% divorced, and 1.3% were widowed.

Additionally, all geographical locations had a good representation; the highest percentage of participants were from the Riyadh region 26.5%, 18.7% from the Asir region, 16.9% from Makkah, 9.7% from the Eastern region, and 5% from Madinah. Moreover, 4.5% of participants were from Jazan, and 4.2% were from Najran. There were other participants from Tabuk (3.4%), Qassim (3.1%), and Al Baha (2.6%). These descriptors are similar to those of the overall population. Regarding the living status of participants, the majority lived with family (86.1%), while 13.9% lived alone.

Regarding the education of participants, 52.0% were college graduates, 24.0% had attended high school, 7.9% had a Diploma, 4.3% had a master’s degree, 4.1% had middle school education, and 3.7% had elementary education. However, 2.3% of participants were not educated and 1.6% had a Ph.D. education. Regarding employment status, the majority (42.9%) had occupations such as being physicians, students, accountants, soldiers, etc., while 18.8% were not employed, and 9.6% were teachers, 7.4% were government employees, 5.9% were health care workers, and 4.3% were private employees, 3.4% engineers, and 1.6% businessmen.

In addition, a high proportion (27.3%) had 11,000–15,000 SR as their monthly income. However, 18.0% of participants had low monthly income <5000 SR. However, 6.8% of participants had monthly income over 25,000 SR (Table 1).

As shown in Table 2, Nearly 106 (17.1%) participants had a chronic health problem/comorbidity. As for COVID infection, 178 (28.7%) of the study participants reported having COVID-19 infection. The COVID-19 vaccine was received among 587 (94.7%) of the study respondents, of whom, 134 (22.8%) received only one dose and 452 (77.2%) received the two doses. Among those who received only one dose, 106 (79.1%) planned to have the second dose. In addition, 334 (53.8%) reported that they will have a third dose if an optional third booster dose is approved. In addition, 351 (56.6%) reported that they were concerned about COVID-19 disease for their own self or family, 436 (70.3%) had family or relatives that had the COVID-19 infection, and 125 (20.3%) lost a family member or relative due to COVID-19 infection.

As shown in Table 3, the demographic variables such as age, gender, geographic location, education, etc. showed no statistically significant association with participants’ attitude towards children’s vaccination (*p* > 0.05).

As shown in Table 4, it was found that there was a statistically significant association between having taken the COVID-19 vaccine (either taking one or two doses) or not being vaccinated, and attitude towards children’s vaccination. The participants who took one dose were significantly higher (29.1%) among non-supporters of children’s vaccination, while those who took the two doses (79.6%) were significantly higher among supporters of children’s vaccination. As well, unvaccinated participants were more common among non-vaccine supporters (*p* < 0.001).

As well, there was a statistically significant association between participants’ attitudes towards vaccinating themselves, and their attitudes towards children’s vaccination. The participants who reported never getting the vaccine were significantly higher among non-supporters of children’s vaccination, whereas those who reported that they were satisfied by taking the first dose, and those who reported taking the two doses, were significantly higher among supporters of children’s vaccination (*p* <0.05) (Figure 1 and Figure 2).

As shown in Table 5, statistically significant associations were found between participants’ attitudes and children’s vaccination (*p* <0.001). False religious beliefs were found to significantly (*p* < 0.001) influence the opposing attitude towards children’s vaccination. As well, statistically significant associations were found between COVID-19 as a health problem, and participants’ attitudes towards children’s vaccination (*p* <0.001). Nearly three-fourths of children’s vaccination supporters strongly believe that COVID-19 is dangerous to their health, while about one-fourth of non-supporters strongly agree that COVID-19 is not dangerous to their health.

As shown in Table 6, marital status and occupation were found to be a significant predictor of children’s vaccination opposition in the study sample.

## 4. Discussion

Vaccination is the appropriate strategy to be applied in a country to help people return to their normal life, and especially to enable children to return to school. Vaccination against COVID-19 in children can also reduce the severity of COVID-19 and multisystem inflammatory syndrome among pediatric cases [19].

There was no statistically significant association between gender and participants’ attitude towards children’s vaccination in our study. A contrary finding has been reported by a recent scoping review [20]. Similarly, gender does not seem to have a statistical significance on parental acceptability of the COVID-19 vaccine for their children in prior studies from Saudi Arabia [21,22].

Our study reports that there was a statistically significant association between participants’ attitudes towards vaccinating themselves, and their attitudes towards children’s vaccination. The participants who reported never getting the vaccine were significantly higher among non-supporters of children’s vaccination, whereas the percentage of support was positively correlated with the number of received doses among the vaccinated participants (*p* <0.05). This is similar to the findings of a recent study from Saudi Arabia where parents who intended to vaccinate themselves (OR: 0.599, 95% CI: 0.367–0.980) and who trust the healthcare system (OR: 0.527, 95% CI: 0.327–0.848) reported greater acceptance of children’s vaccination [18]. In our previous study, the majority of parents strongly agreed (23.4%) and agreed (35.1%) about the importance of getting their children vaccinated, 22.1% of parents strongly agreed, and 33.3% agreed regarding their willingness to get their children vaccinated to prevent coronavirus disease [23]. Parents who had the COVID-19 vaccine were about five-fold more likely to vaccinate their children compared with parents who did not receive the vaccine (OR = 4.9, CI: 3.12–7.70) in a study from EMR [24].

Moreover, a statistically significant association was found between participants’ attitudes towards the third dose, and children’s vaccination. A higher percentage of children’s vaccination supporters (89%) stated that they would get the third dose, while a higher percentage of non-children’s vaccination supporters (38.1%) stated that they would not get the third dose. In another recent study from the Kingdom, ninety percent of the participants agreed that it was essential for everyone to receive the recommended vaccines with their children, 92% believed that vaccines are safe for their children, 91% of the participants agreed to give their new children all the recommended doses, 86% welcomed mass/school vaccination campaigns, and 81% were willing to pay for additional vaccines for themselves and their children [13]. 

No statistically significant associations were found between those having co-morbidities or being infected with COVID-19 before, and their attitude towards children’s vaccination in our study. Intention to take the COVID-19 vaccine was low among patients with chronic diseases, to achieve herd immunity, in a Nigerian study [22]. Another study from Saudi Arabia showed that the smoking status of the parents, having an allergy, and having other comorbidities were risk factors of having persistent post-COVID-19 symptoms (*p* ≤ 0.05) among child age groups [23].

Vaccine newness as a reason for hesitating to get vaccinated was the most reported by the non-vaccine supporters in this study. This is similar to the findings of Sultan et al. where hesitancy was mainly driven by the novelty of the vaccines and the fear of serious adverse effects [25]. Vaccine hesitancy is defined as a delay in the acceptance or refusal of vaccines, despite the availability of vaccination services to the public. Many factors influence vaccine hesitation such as timing, place, and type of vaccine. Various elements, such as complacency, convenience, and confidence also influence vaccine hesitancy development [11]. Studies conducted in the Kingdom have reported a variable level of vaccine hesitancy, between 20–72%, among parents, in regard to immunizing their children against COVID-19 [14,26,27,28,29].

Statistically significant associations were found between participants’ attitudes and children’s vaccination in the present study. Most parents reported their concerns about the side effects of the COVID-19 vaccine (*n* = 108, 60%). However, still, about 40% of the parents reported that they are very excited about the vaccination and believe that it will be protective (*n* = 76) in a Turkish study [30].

False religious beliefs were found to significantly influence the opposing attitude towards children’s vaccination. As well, statistically significant associations were found between COVID-19 as a health problem and participants’ attitude towards children’s vaccination. Similarly, parents who perceived low benefit from the vaccine (OR = 16.3; 95% CI, 12.1–21.9) or who had safety or efficacy concerns (OR = 3.76; 95% CI, 3.10–4.58) were among the most hesitant to vaccinate their children [29]. Vaccine safety and efficacy were parents’ top concerns, and receiving more information about safety and efficacy were the top facilitators of COVID-19 vaccination for this age group in a US Study [31]. In another US study, only 31.3% of parents intended to vaccinate their child, 22.6% were unsure, and 46.2% intended not to vaccinate. Logistic regression indicated significant barriers to vaccination uptake, including concerns about immediate and long-term vaccination side effects for young children, the rushed nature of FDA approval and distrust in government and pharmaceutical companies, lack of community and family support for pediatric vaccination, conflicting media messaging, and lower socioeconomic status [32]. Understanding which individuals are vaccine-hesitant and why, what barriers to accessing vaccination services exist, and how to cultivate vaccine confidence is essential to inform the development of tailored strategies to increase vaccine uptake [33]. Our study includes data from different geographical areas inside the Kingdom, and an extensive analysis has been made which is the strength of our study. 

However, the following factors may have led to certain limitations in the present study. The cross-sectional design of this study cannot confirm the causality of the relationship between compared variables. The self-reported response could over or underestimate the results.

## 5. Conclusions

Health professionals and policy makers should implement and support strategies to ensure children are vaccinated against COVID-19. They also need to educate parents and families regarding the importance of vaccination against COVID-19. This can go a long way in reducing the impact of hesitancy, and fear, among them. The study might help to design interventions to address the impediments in the current COVID-19 vaccination drives and vaccinate a wider child population to limit the pandemic.

## Figures and Tables

**Figure 1 vaccines-11-00250-f001:**
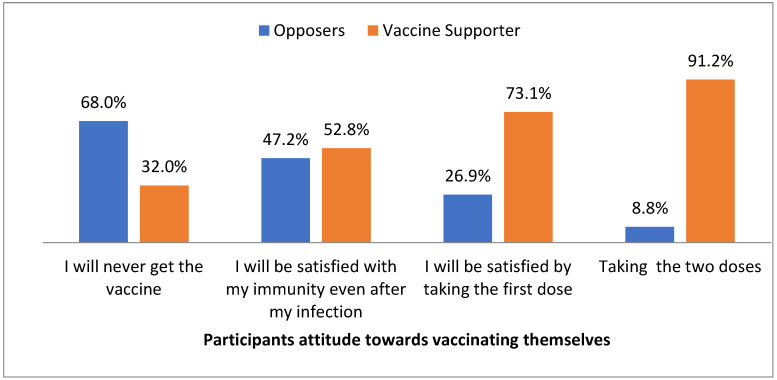
Participants’ attitudes towards vaccinating themselves.

**Figure 2 vaccines-11-00250-f002:**
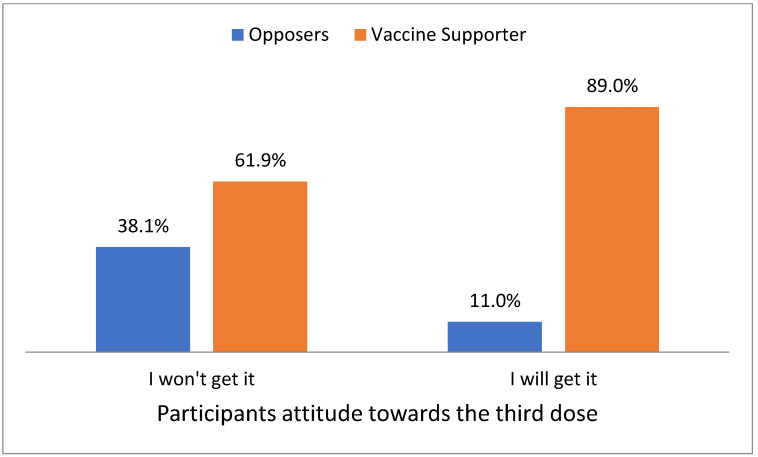
Participants’ attitudes towards the third dose.

**Table 1 vaccines-11-00250-t001:** Socio-demographics characteristics of participants (*n* = 620).

Variables	No.	Percentage
Gender		
Male	355	57.2
Female	265	42.8
Age (years)		
18–25	211	34.0
26–35	186	30.0
36–40	67	10.8
41–45	62	10.0
46–50	46	7.4
>50	48	7.8
Geographical location		
Riyadh	164	26.5
Asir	116	18.7
Makkah	105	16.9
Eastern Region	60	9.7
Medina	31	5.0
Jazan	28	4.5
Najran	26	4.2
Tabuk	21	3.4
Qassim	19	3.1
Al Baha	16	2.6
Northern borders	13	2.1
Hail	11	1.8
Al Jouf	10	1.6
Education		
Not educated	14	2.3
Elementary	23	3.7
Middle school	25	4.1
High school	149	24.0
Diploma	49	7.9
College Graduate	323	52.0
Master	27	4.3
PhD	10	1.7
Occupation		
Teacher	59	9.6
Engineer	21	3.4
Healthcare worker	37	5.9
Government employee	46	7.4
Private employee	27	4.3
An employee in a commercial private sector	38	6.1
Unemployed	116	18.8
Businessman	10	1.6
Other	266	42.9
Monthly household income		
<5000	112	18.0
5000–10,000	133	21.5
11,000–15,000	169	27.3
16,000–20,000	112	18.0
21,000–25,000	52	8.4
>25,000	42	6.8
Marital status		
Single	252	40.6
Married	329	53.1
Divorced	31	5.0
Widow	8	1.3
Living status		
With family	534	86.1
Alone	86	13.9

**Table 2 vaccines-11-00250-t002:** COVID-19 infection and vaccination data among study participants.

Variables	No.	Percentage
Comorbidities among sample		
Yes	106	17.1
No	514	82.9
COVID-19 infection		
Yes	178	28.7
No	442	71.3
COVID-19 vaccination history		
Yes	586	94.7
No	33	5.3
1st dose	134	22.8
2nd dose	452	77.2
Concerns about COVID-19 disease for own self or family		
Yes	351	56.6
No	269	43.4
COVID-19 infection among participants’ families or relatives		
Yes	436	70.3
No	184	29.7
Loss of a family member or relative due to infection with COVID-19		
Yes	125	20.1
No	495	79.9
Planning to take the second dose		
Yes	106	79.1
No	28	20.9
Third dose of the vaccine		
I won’t get it	286	46.2
I will get it	334	53.8

**Table 3 vaccines-11-00250-t003:** Relationship between socio-demographic variables and participants’ attitudes towards children’s vaccination.

Demographics Variables	Categories	Children Vaccination		*p*-Value
		Opposers	Vaccine Supporter	
		*n* (%)	*n* (%)	
Age	18–25	50 (23.6)	161 (76.4)	0.688
	26–35	54(29.0)	132 (71.0)	
	36–40	15(22.4)	52 (77.6)	
	41–45	16 (25.8)	46 (74.2)	
	46–50	9 (19.6)	37 (80.4)	
	>50	11 (22.9)	37 (77.1)	
Gender	Male	80 (22.5)	275 (77.5)	0.124
	Female	74 (27.9)	191 (72.1)	
Geographic Location	Riyadh	47 (28.7)	117 (71.3)	0.533
	Qassim	4 (21.1)	15 (78.9)	
	Eastern Region	11 (18.3	49 (81.7)	
	Makkah	23 (21.9)	82 (78.1)	
	Medina	7 (22.6)	24 (77.4)	
	Hail	1 (9.1)	10 (90.9)	
	Al Jouf	2 (20.0)	8 (80.0)	
	Tabuk	3 (14.3)	18 (85.7)	
	Northern borders	5 (38.5)	8 (61.5)	
	Asir	33 (28.4)	83 (71.6)	
	Jazan	7 (25.0)	21 (75.0)	
	Najran	4 (15.4)	22 (84.6)	
	Al Baha	6 (37.5)	10 (62.5)	
Education	Not educated	3 (21.4)	11 (78.6)	0.897
	Elementary	7 (30.4)	16 (69.6)	
	Middle school	9 (36.0)	16 (64.0)	
	High school	37 (24.8)	112 (75.2)	
	Diploma	11 (22.4)	38 (77.6)	
	College graduate	76 (23.5)	247 (76.5)	
	Master	7 (25.9)	20 (74.1)	
	PhD	3 (30.0)	7 (70.0)	
Occupation	Teacher	16 (27.1)	43 (72.9)	0.054
	Engineer	3 (14.3)	18 (85.7)	
	Health care worker	6 (16.2)	31 (83.8)	
	Government employee	9 (19.6)	37 (80.4)	
	Private employee	7 (25.9)	20 (74.1)	
	Employee in a commercial private sector	12 (31.6)	26 (68.4)	
	Unemployed	40 (34.5)	76 (65.5)	
	Businessman	-	10 (100.0)	
	other	63 (23.6)	203 (76.4)	
Monthly household income	<5000	35 (31.3)	77 (68.8)	0.183
	5000–10,000	32 (24.1)	101 (75.9)	
	11,000–15,000	48 (28.4)	121 (71.6)	
	16,000–20,000	21 (21.4)	77 (78.6)	
	20,001 and above	20 (17.8)	92 (82.2)	
Marital status	Single	55 (21.9)	197 (78.1)	0.059
	Married	85 (25.8)	244 (74.2)	
	Divorced	7 (22.6)	24 (77.4)	
	Widow	5 (62.5)	3 (37.5)	
Living status	With family	136 (25.4%)	398 (74.6%)	0.255
	Alone	17 (19.8%)	69 (80.2%)	

**Table 4 vaccines-11-00250-t004:** Association between factors related to COVID-19 and participants’ attitudes towards children’s vaccination.

Factors	Categories	Children Vaccination		*p*-Value
		Opposers	Vaccine Supporter	
		*n* (%)	*n* (%)	
Co-morbidities	Yes	24(22.6)	82 (77.4)	0.593
	No	129 (25.1)	385 (74.9)	
Previous COVID-19 infection	Yes	39 (21.9)	139 (78.1)	0.310
	No	114 (25.8)	328 (74.2)	
COVID-19 Vaccination history	1st dose	39 (29.1)	95 (70.9)	<0.001 **
	2nd dose	92 (20.4)	360 (79.6)	
	Not vaccinated	20 (60.6)	13 (39.4)	

** Highly significant.

**Table 5 vaccines-11-00250-t005:** Participants’ attitudes, beliefs, and emotions towards COVID-19 vaccination among children.

Attitudes Items	Categories	Children Vaccination		*p*-Value
		Opposers	Vaccine Supporter	
		*n* (%)	*n* (%)	
Vaccination can prevent COVID-19	Strongly agree	3 (12.0)	22(88.0)	
	Agree	45 (13.0)	302 (87.0)	<0.001 **
	Neither	24 (25.5)	70 (74.5)	
	Disagree	52 (44.4)	65 (55.6)	
	Strongly disagree	17 (45.9)	20 (54.1)	
The vaccine can prevent me from being a spreader	Strongly agree	2 (8.0)	23(92.0)	
	Agree	33 (13.9)	204 (86.1)	<0.001 **
	Neither	38 (20.7)	146 (79.3)	
	Disagree	50 (36.2)	88 (63.8)	
	Strongly disagree	16 (44.4)	20 (55.6)	
Relying on faith alone without any action to prevent infection.	Strongly agree	28 (51.9)	26 (48.1)	<0.001 **
	Agree	18 (40.0)	27 (60.0)	
	Neither	23 (33.3)	46 (66.7)	
	Disagree	49 (20.7)	188 (79.0)	
	Strongly disagree	34 (15.8)	181 (84.2)	
Social media has a role in increasing misconceptions and incorrect information about the vaccine.	Strongly agree	42(29.2)	102(70.8)	0.519
	Agree	24(22.6)	82(77.4)	
	Neither	20(26.7)	55(73.3)	
	Disagree	29(20.6)	112(79.4)	
	Strongly disagree	37(24.0)	117(76.0)	
I think COVID-19 is not dangerous to my health	Strongly agree	13(26.5)	36(73.5)	<0.001 **
	Agree	32(41.6)	45(58.4)	
	Neither	30(28.0)	77(72.0)	
	Disagree	39(19.5)	160(80.5)	
	Strongly disagree	36(19.1)	152(80.9)	

** Highly significant.

**Table 6 vaccines-11-00250-t006:** Predictors of the children vaccination opposition using multivariate logistic regression analysis.

Variables		OR	95% CI
Marital Status	Married (ref.)	1	
	Un married	113.17	17.84–65.55
Employment Status	Employed (ref.)	1	
	Un Employed	19.14	0.109–0.43

## Data Availability

Not applicable.

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
