# Peer review of "Community Perception and Attitude towards COVID-19 Vaccination for Children in Saudi Arabia"

_vaccines, 2023, doi:10.3390/vaccines11020250_

Round 1

Reviewer 1 Report

This is an interesting article on Community Perception and Attitude towards COVID-19 Vaccination for Children in Saudi Arabia.

Introduction is essential.

Materials and methods are clear and detailed.

Results are properly presented with the help of tables. Adequate statistical analysis is also provided.

The results are adequately discussed with reference to existing literature and articles dealing with similar subject.

 Minor changes are indicated in the text.

Author Response

As suggested by the respected reviewer, the suggested changes in the text have been done in track changes.. Please see the revised manuscript.

Poster dose has been corrected to booster dose. (Please see Page 6, lines 186)

References have been removed before full stop. (Please see Page 11, lines 297)

Reference 22 has been removed as it was cited twice.

Reviewer 2 Report

This is an interesting and important topic regarding the attitudes of adults in relation to childhood COVID-19 vaccination in Saudi Arabia. However, the language requires a lot of improvement throughout and the methods are not appropriately detailed for publishing at this time. There also appear to be some errors in the results. Please find my specific comments below:

Abstract:

“This study was undertaken to find out the percentage of supporters and opposers in the Saudi population…”. Aim seems repetitive.

 “cross-sectional model with complete randomization” I believe this statement is incorrect. This appears to be a cross sectional survey, not model. 

“Data were analyzed using statistical software IBM SPSS version” how were data analysed is more important to explain here than software used.

Introduction

Language needs to be improved throughout. 

Need to explain which vaccines are recommended and funded in Saudi Arabia.

Vaccine hesitancy is low- what does low mean, need to be specific. Does positive attitude mean intent to vaccinate?

“Some studies in the Kingdom of Saudi Arabia 47 have concluded that vaccine hesitancy towards COVID-19 among parents is a concern 48 and is likely to influence the COVID-19 vaccination”. Need better linking for this sentence. You seem to be saying that adult vaccine hesitancy impacts childhood uptake, which is true, but earlier you said vaccine hesitancy in adults is low. So need to be clear about the problem.

You also discuss comorbidities in the results and discussion. Why was this included. Needs to be introduced here.

Methods

Methods need more detail throughout.

“Cross-sectional model with complete randomization of participants nationwide”. I don’t know what this means, there was no intervention. Did you have a list of every mobile phone user and the list of those you invited to participant was a random sample from the original list. This sounds like a cross sectional survey design. Needs clarification.

“The study included adults residing in different geographical locations of Saudi Ara-68 who consented to participate in the study and were enrolled in the study's final analysis”. but how did you identify these people to start with?

Sample size, “to detect a single proportion with…”. A single proportion of what?

“Was published in the course”, what course?

How was the questionnaire developed? Was it based on previous literature/expert opinion? Is it validated?

“it was revised, coded, and fed to statistical software”. Very vague

Data analysis description could be better explained. It sounds like you used descriptive analysis and then looked for associations between vaccine uptake and demographic variables but it isn’t very clear.

Results

What was the response rate, i.e. how many people were invited.

“All geographical locations had a good representation” what does good representation mean. Are these descriptors similar to that of the overall population?

Table 2- comorbidities – how were these defined?

How were vaccine opposers and supporters defined?

Remove table 5. Numbers are too small to include in analysis.

I believe the incorrect proprotions are shown in a number of tables (i.e. table 6). This should show column proportions not row proportions. This suggests the analysis for these sections also needs to be rechecked.

Discussion

Overall, the discussion includes most of the main points but the language and structure needs improving. The authors should also focus more on what this means and how this information could be useful. 

“Interestingly, in another study, one woman was worried about her 238 husband who suffers from a chronic disease. She reported her great willingness to vaccinate her husband and her family members to protect him” remove, based on one person

Statistically significant associations were found between reasons for wanting to get vaccinated among the non-vaccinated study population, and participants' attitudes towards children vaccination in the present study.- I’m not sure what you mean by this

“A support program for health education for these parents should be introduced”. Why? This has not been mentioned previously. This does not link to the results well.

Author Response

Reviewer 2

As suggested by the respected reviewer the following corrections have been made.

Abstract:

“This study was undertaken to find out the percentage of supporters and opposers in the Saudi population…”. Aim seems repetitive.

The word opposers has been removed. Please see Page 1, lines 13

 “cross-sectional model with complete randomization” I believe this statement is incorrect. This appears to be a cross sectional survey, not model. 

The word model has been removed. Please see Page 1, lines 16

“Data were analyzed using statistical software IBM SPSS version” how were data analysed is more important to explain here than software used.

As suggested this has been revised. Please see Page 1, lines 18-19

Introduction

Language needs to be improved throughout. 

As suggested Language has been improved with the help of grammarly software.

Need to explain which vaccines are recommended and funded in Saudi Arabia.

Done, Please see lines 44-45

Vaccine hesitancy is low- what does low mean, need to be specific. Does positive attitude mean intent to vaccinate?

“Some studies in the Kingdom of Saudi Arabia 47 have concluded that vaccine hesitancy towards COVID-19 among parents is a concern 48 and is likely to influence the COVID-19 vaccination”. Need better linking for this sentence. You seem to be saying that adult vaccine hesitancy impacts childhood uptake, which is true, but earlier you said vaccine hesitancy in adults is low. So need to be clear about the problem.

Done, Please see lines 48-52

You also discuss comorbidities in the results and discussion. Why was this included. Needs to be introduced here.

 Done, Please see lines 58-66

Methods

Methods need more detail throughout.

“Cross-sectional model with complete randomization of participants nationwide”. I don’t know what this means, there was no intervention. Did you have a list of every mobile phone user and the list of those you invited to participant was a random sample from the original list. This sounds like a cross sectional survey design. Needs clarification.

 This has been clarified, Please see lines 74-77

“The study included adults residing in different geographical locations of Saudi Ara-68 who consented to participate in the study and were enrolled in the study's final analysis”. but how did you identify these people to start with?

 This has been modified, Please see lines 87

Sample size, “to detect a single proportion with…”. A single proportion of what?

This has been modified, Please see lines 91

“Was published in the course”, what course?

The word course has been removed, please see lines 97

How was the questionnaire developed? Was it based on previous literature/expert opinion? Is it validated?

This has been modified, Please see lines 99-111

“it was revised, coded, and fed to statistical software”. Very vague

 This has been modified, Please see lines 137-140

Data analysis description could be better explained. It sounds like you used descriptive analysis and then looked for associations between vaccine uptake and demographic variables but it isn’t very clear.

 This has been modified, Please see lines 142-143

Results

What was the response rate, i.e. how many people were invited.

Response rate is 92.6%. 670 people were invited Please see line 93

“All geographical locations had a good representation” what does good representation mean. Are these descriptors similar to that of the overall population?

As suggested, this has been mentioned, Please see lines 169 -170

Table 2- comorbidities – how were these defined?

The comorbidities were defined as generally any chronic diseases. Please see lines 121 -127

How were vaccine opposers and supporters defined?

Vaccination supporter are those who answered yes for children vaccination, and the opposers are those who said no for children vaccination. Please see lines 136 -137

Remove table 5. Numbers are too small to include in analysis.

As suggested table 5 has been removed.

I believe the incorrect proprotions are shown in a number of tables (i.e. table 6). This should show column proportions not row proportions. This suggests the analysis for these sections also needs to be rechecked.

 As suggested the analysis has been rechecked and required modifications have been made.

Discussion

Overall, the discussion includes most of the main points but the language and structure needs improving. The authors should also focus more on what this means and how this information could be useful. 

The discussion has been improved (Please see 285-290)

“Interestingly, in another study, one woman was worried about her 238 husband who suffers from a chronic disease. She reported her great willingness to vaccinate her husband and her family members to protect him” remove, based on one person

As suggested this has been removed (Please see 283-285)

Statistically significant associations were found between reasons for wanting to get vaccinated among the non-vaccinated study population, and participants' attitudes towards children vaccination in the present study.- I’m not sure what you mean by this

As suggested this has been revised (Please see 301-303)

“A support program for health education for these parents should be introduced”. Why? This has not been mentioned previously. This does not link to the results well.

As suggested this has been removed  (Please see 332-333)

Thank you